# Estimating age of mule deer in the field: Can we move beyond broad age categories?

**Morgan S. Hinton**[1]* , **Brock R. McMillan**[1] , **Kent R. Hersey**[2] , **Randy T. Larsen**[1]

**1** Department of Plant and Wildlife Sciences, Brigham Young University, Provo, UT, United States of America, **2** Utah Division of Wildlife Resources, Salt Lake City, UT, United States of America

☙ These authors contributed equally to this work.
* morganshay527@gmail.com

## Abstract

Age of individuals is an intrinsic demographic parameter used in the modeling and management of wildlife. Although analysis of cementum annuli from teeth is currently the most accurate method used to age ungulates, the age of live ungulates in the field can be estimated by examining tooth wear and tooth replacement patterns. However, there may be limitations to aging based on tooth wear as the rate of tooth wear likely varies among individuals due to factors such as age, diet, environment, and sex. Our objective was to determine the reliability of estimating age for mule deer based on tooth wear and tooth replacement patterns. We compared ages estimated by tooth wear (collected at time of capture for a statewide monitoring effort) to ages determined from cementum analysis (from teeth collected after mortalities of radio-tracked animals from the monitoring effort). Accuracy was high; ages estimated from tooth wear were within one year of cementum ages >75% of the time when aged by experienced observers. Bias in accuracy for estimates of age was low but slightly biased toward underestimation (i.e., 0.6 years on average)—especially as cementum age increased. Our results indicate that aging mule deer using patterns in tooth wear can be reliable if observers estimating age have experience using this method.

## Introduction

Accurate estimates of demographic rates are essential for modeling populations and managing wildlife [1]. The collection and analysis of reliable data can provide detailed information on parameters of a population such as recruitment, reproduction, sex ratios, and survival [2–4]. These parameters can inform models of population change by indicating increases, decreases, or constant rates associated with each parameter, thereby improving understanding of the drivers of population dynamics [5]. Additionally, wildlife managers rely on accurate data and models to inform management decisions such as determining the number of annual hunting permits [6, 7].

Demographic rates can vary notably across ages and by sex for individuals in a population [8]. Whereas sex can typically be easily determined in the field for most large mammal species, age is often more difficult to estimate. Age structure of populations, however, can have strong influences on demographic parameters that may lead to fluctuations in population size [9, 10].

www.mammalsociety.org/), Brigham Young University (https://www.byu.edu/), the Bureau of Land Management (https://www.blm.gov/), the Mule Deer Foundation (https://muledeer.org/), the Rocky Mountain Elk Foundation (https://www.rmef.org/), Safari Club International (https://safariclub.org/), Sportsmen for Fish and Wildlife (https://sfw.net/), the Utah Archery Association (https://www.utaharchery.org/), and the Utah Division of Wildlife Resources (https://wildlife.utah.gov/). Grant numbers 196313 and 206012 from Utah Division of Wildlife Resources were awarded to BRM and RTL. Additionally, the Utah Division of Wildlife Resources contributed to the data collection associated with this research. All other funders had no role in study design, data collection and analysis, decision to publish, or preparation of the manuscript.

**Competing interests:** The authors have declared that no competing interests exist.

Further, age influences resistance to disease, fertility, litter size, growth rate of neonates, body size, resource selection, and patterns of movement [9, 11–14]. Thus understanding the age structure of a population is helpful for conservation and management. Estimation of age structure, however, can only be determined by collecting accurate data on age from a sufficient sample of individuals within the population.

Estimating age of large mammals in the field is most commonly done by evaluating dental characteristics [15–19]. For example, the first premolar in the lower or upper jaw is frequently extracted from live carnivores to more precisely determine age using cementum analysis [20]. Tooth extraction for other live large mammals including ungulates is considerably more invasive than the procedure for carnivores and is generally avoided due to the potential for reduced food intake or risk of damaging adjacent teeth during the extraction process [21]. In place of tooth extraction, patterns of tooth replacement and wear in ungulates can be used to estimate age of ungulates between 0.5 and 2.5 years of age, but the use of replacement patterns can be unreliable beyond 2.5 years of age due to the permanent formation of dental structure [19, 22, 23]. After formation of permanent teeth, degree of wear on molars and incisors can be used to estimate ages of individuals [24, 25].

Due to variation in patterns of tooth wear and difficulty in accurate estimation of age for cervids, it is common practice to place individuals into age categories such as fawn, yearling, and adult or to estimate age to 2.5 years using patterns of replacement, but then place older individuals into a category of ≥3.5 years [26, 27]. While classifying individuals into broad age categories may limit risk of inaccurately aging individuals, it does not provide detailed information regarding the age structure of a population or how age may influence demographic rates. Specifically, this practice lacks the ability to parse out older animals near the end of expected lifespans where demographic rates are generally believed to diminish with senescence [28]. Improvement over these broad categories when estimating age for live animals would enhance our understanding of cervid ecology and better inform their conservation and management. However, estimation of actual age for many ungulates is generally considered inaccurate and therefore, unreliable.

Because accurate estimation of age of cervids is fundamental to all aspects of their population ecology, our objective was to determine if estimation of actual age from experienced biologists using patterns of tooth wear could provide reliable data that would be more informative than using broad categories of age as currently recommended. Specifically, we determined the accuracy of age estimation using patterns of tooth replacement and tooth wear for mule deer (*Odocoileus hemionus*), a species of great social and biological interest in western North America. We predicted that the use of tooth replacement and wear to estimate age could be reliable if observers had adequate training and experience with this technique. We also predicted that as age of deer increased, accuracy of age estimation would decrease due to greater variation between individuals in tooth wear for older animals. The results of this study will help determine if using patterns in tooth replacement and wear can be a reliable method to estimate the age of mule deer beyond broad age categories.

## Study area

Our study was conducted across the state of Utah, comprised of mule deer captured from 21 management units within five administrative regions of the Utah Division of Wildlife Resources (Fig 1). Latitude ranged from 37˚N to 42˚N and longitude ranged from 109˚W to 114˚W. Elevation in Utah ranged from 762 meters above sea level in Southwestern Utah to 4114 meters above sea level at King's Peak in the Uinta mountain range located in Northeastern Utah. Annual precipitation varied from ≤10 centimeters in the deserts to 150 centimeters

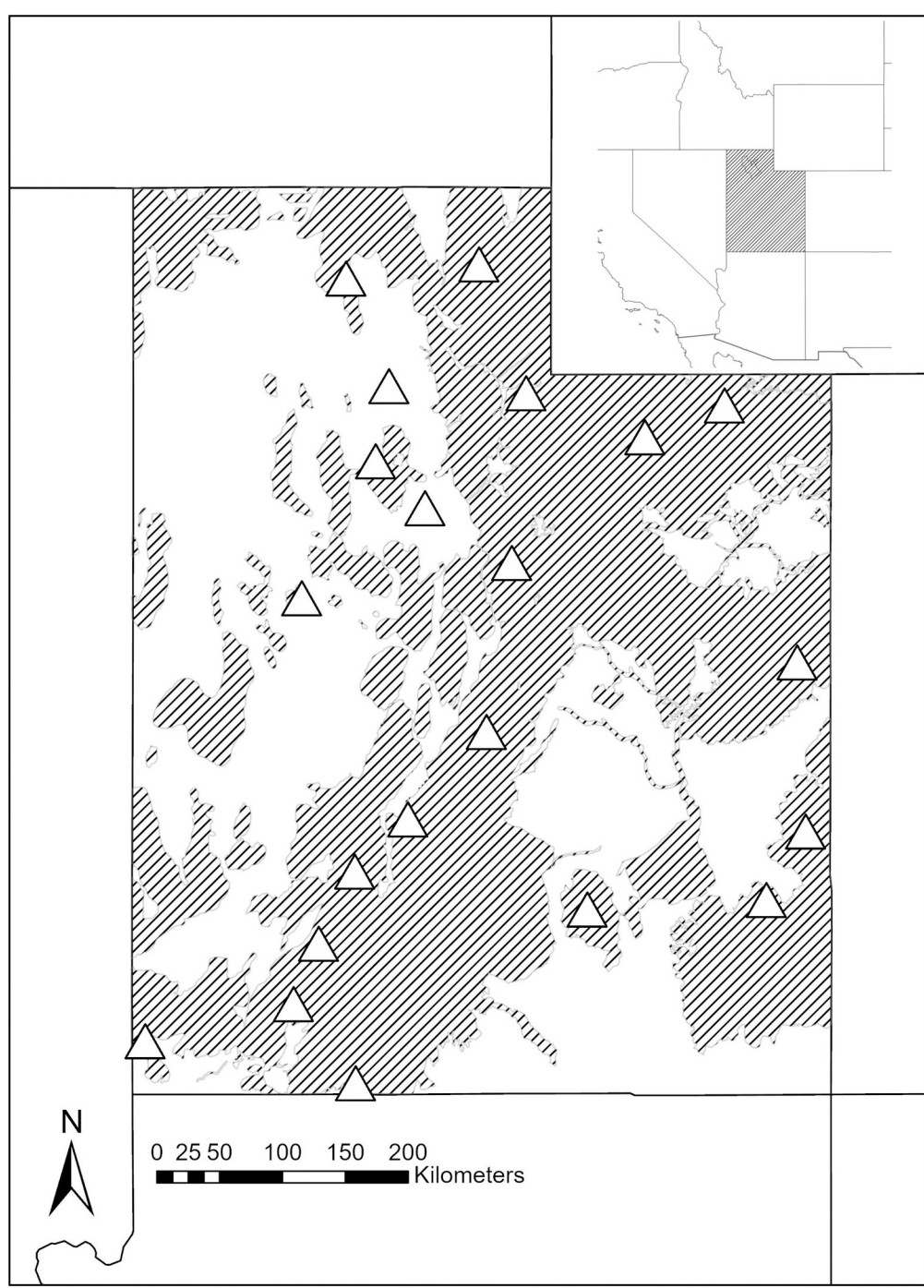

**Fig 1. Locations (marked by Δ) within Utah, USA where we captured mule deer (*Odocoileus hemionus*) from 2014–2020 and estimated age via tooth replacement and wear.** Mule deer habitat denoted by shading. Map base layer can be found at https://utah.maps.arcgis.com/home/item.html?id=543fa1f073714198a3dbf8a292bdf30c.

in higher mountain areas and was a mix of rain and snow [29]. Further, sampling occurred in seven level III ecoregions including alpine, desert, irrigated valley, mountain forest, riparian, shrubland, and woodland habitat types [30]. Generalized vegetative communities within these

ecoregions frequently inhabited by mule deer included the mountain brush zone, pinyon-juniper woodlands, sagebrush steppe, and subalpine zone [31]. Other ungulate species that occured with mule deer in Utah included bighorn sheep (*Ovis canadensis*), bison (*Bison bison*), moose (*Alces alces*), mountain goats (*Oreamnos americanus*), pronghorn (*Antilocapra americana*), Rocky Mountain elk (*Cervus canadensis*), feral horses (*Equus caballus*), and a variety of domestic livestock. Predators of mule deer in Utah included bobcats (*Lnyx rufus*), black bears (*Ursus americanus*), cougars (*Puma concolor*), and coyotes (*Canis latrans*).

## Methods

### Capture and estimation of age

During late November through early March 2014–2020 in conjunction with the Utah Division of Wildlife Resources, we captured adult mule deer via helicopter net-gunning [32–35]. Deer were transported in transport bags by helicopter after capture to a central landing zone where a ground crew comprised of biologists processed the animals. All capture and collaring protocols for mule deer were approved by the Brigham Young University Institutional Animal Care and Use Committee (IACUC) (protocol 150110) and consistent with the published guidelines for use of wild animals in research by the American Society of Mammalogists [36]. We fitted each individual with a GPS tracking collar programmed with a mortality sensor to alert us in the event of a study animal mortality (model type G2110E2H, G5-2DH, or W300; Advanced Telemetry Systems, Isanti, MN, USA). We also collected basic biometric data in addition to age including several body measurements, body condition score, rump fat depth, loin thickness, lactation score, body mass, heart rate, and respiration rate prior to release [37–39]. Following collection of biological data and fitting with a collar, deer were released for remote monitoring via GPS collars.

Age of each individual was estimated by experienced biologists using patterns in tooth replacement and wear for both molars and incisors [24, 25, 40]. Age was estimated using the same criteria for both males and females. We considered biologists that had previously aged over 200 individual mule deer of both sexes across all capture units to be experienced observers. We chose 200 deer as a cutoff because observers had been exposed to many deer of both sexes and of all ages, especially older individuals by that point in time. Fawns were aged in the field by the capture crew using patterns of tooth replacement and presence of temporary milk teeth. We used tooth replacement patterns to identify deer 1.5 years old to older than 1.5 years. Adult mule deer follow the dental formula (incisor/canine/premolar/molar) of 0/3, 0/1, 3/3, 3/3, making younger animals easier to identify by lack of permanent tooth eruption. Fawns under one year of age typically have three to four fully erupted teeth along each side of the jaw including three premolars and occasionally a single molar, often referred to as "milk teeth" since they are present at birth and are eventually replaced by permanent teeth. Yearling deer are characterized by three cusps on the third premolar and only partial eruptions of the third molar. Two-year-old deer have noticeably sharp lingual crests on their third premolar and first molar, as well as the eruption of the fourth and final dental formula. For animals ≥ 3 years of age, we used tooth wear to estimate age of individuals. This method uses both wear found on incisors and lingual crests on molars as well as degree of staining on lingual crests and around the gum line to estimate age. Shorter crest heights and higher degree of staining indicate progression in age. See Table 1 for general dental characteristics used to estimate ages of mule deer. These characteristics are not absolutes, but rather general guidelines that have been supported by frequent feedback determined from cementum analysis.

**Table 1. Dental characteristics associated with ages (in years) of free-ranging mule deer (*Odocoileus hemionus*) in western North America.**

| Age | Incisor Wear | Molar Wear |
|---|---|---|
| 1.5 | No wear on incisors Deciduous teeth often present or permanent teeth only partially erupted | No wear on molars. P3 with 3 cusps and partial eruption of M3 |
| 2.5 | Virtually no wear on incisors | Thin lines of dentin on lingual crests of molars |
| 3.5 | Virtually no wear on incisors | Virtually no wear on molars with thicker dentin lines than 2.5 year olds |
| 4.5 | Slight wear on the inside of the incisors | Slight wear beginning on crests of M1 |
| 5.5 | Moderate wear on the inside of the incisors, but limited to no "dishing" | Moderate wear beginning on crests of M1 with slight wear beginning on M2 |
| 6.5 | Incisors showing both wear and evidence of starting to dish | Moderate wear on crests of M1and M2 with slight wear beginning on M3 |
| 7.5 | Pronounced wear on the incisors with dishing present | Moderate wear on crests of M2 and M3 Flattening of M1, but tooth is not "dished out" |
| 8.5 | Heavy wear on incisors with dishing and teeth approaching gum line | M1 dished out. Heavy wear or flattening on M2 and M3, but not "dished" |
| 9.5 | Dishing and incisors approaching gum line | M1 dished out. M2 with very heavy wear or dished out. M3 heavy wear |
| 10.5 + | Incisors dished and often completely worn down towards gum line | All molars dished out with older animals having molars approaching gum line |

## Incisor collection and cementum analysis

Upon receiving a mortality notification from GPS-collared mule deer, biologists examined carcasses to determine cause-specific mortality and extracted one, or both, of the I1 incisors from the lower jaw [41, 42]. All incisors were then placed in envelopes labeled with the six-digit collar code of the deer, mortality date, sex, and location. Incisors were sent to the Wildlife Ecology Research Lab at Brigham Young University (www.wildlifeecologylab.byu.edu) where we processed the teeth. Upon receipt, we soaked teeth from one to two weeks depending on size in a decalcifying solution composed of formic acid, formaldehyde, and distilled water. Once softened, we sliced teeth in horizontal cross-sections near the root using a Leica CM1850 cryostat machine (model CM 1850-3-1). We placed tooth cross-sections on glass slides and stained them for 90 seconds using Crystal Violet biological stain solution. Following staining, we examined cross sections under a microscope and counted annuli to determine age. During the cementum analysis, we assigned each age a confidence rating of A, B, or C. We used "A" for estimate of age that were clear given distinct annuli patterns. We used a "B" rating for estimates of age that were believed to be accurate within a range of ± 1 year. We used a "C" rating for ages that were less clear and believed to be accurate within ± 2 or more years. We chose to exclude any ages assigned a "C" rating from our dataset due to uncertainty. Estimation of age from cementum analysis was "blind" to estimates of age from tooth replacement and wear (essentially a 'double-blind' design).

We used analysis of cementum annuli to provide a benchmark for age comparison with estimates collected on live animals in the field. We selected this method because cementum analysis is the most accurate method of age determination for wild ungulates [43, 44]. Cementum is continuously deposited on permanent teeth throughout an animal's lifetime. The rate of cementum deposition coincides with environmental conditions, consistently resulting in a large, light-colored deposition throughout spring, summer, and fall, and a thin dark band (called an annuli) in the winter due to nutritional stress in northern latitudes [44]. In these latitudes, each annular ring represents one year of life, not including the first year of life when animals have temporary adolescent teeth. The annuli may be used to estimate age by cutting,

staining, and viewing cross sections of each tooth under a microscope. Literature suggests cutting teeth closest to the posterior end of the root without taking the root tip to increase accuracy in the aging process [45].

## Data analysis

We compiled a record of deer that had an age assigned to them via cementum analysis following mortality of each individual deer. We compared deer ages determined from cementum analysis with the ages estimated from tooth replacement and wear after correcting for the number of intervening years between capture (wear age) and mortality (cementum age). We used a generalized linear mixed-effects model with a binomial distribution for error structure within the "lme4" package in program R to assess accuracy and evaluate the difference between age determined from cementum with age estimated in the field from tooth replacement and wear [46–48]. When wear age was estimated within plus or minus 1-year of cementum analysis age, we set the response variable to 1 (accurate); when they differed by more than a year, we set the response variable to 0 (inaccurate). We chose within 1-year for accuracy because most estimates of wear on live animals occurred in either December or March, but mortalities and the subsequent age derived from cementum analysis occurred throughout the year.

We identified explanatory variables as cementum age (in years), a binary variable for observer experience (experienced set as 1; inexperienced set as 0), sex of the deer, and period of capture (Fall for animals captured in November and December, Spring for animals captured in February or March). We set capture unit, year, and unique ID (to account for repeated measures) as random effects (intercept only) and included each of them in all models. Unit of capture was included as a random effect because previously analyses indicated it had no significant influence on accuracy of age estimates. Because our analysis was observational in nature as opposed to experimental, we used model selection and an information-theoretical approach [49, 50]. We first formulated a set of 19 *a priori* models with combinations of explanatory variables representing hypotheses about which factors influence accuracy between estimates of age from cementum analysis and assessment based on tooth wear. Before formulating models, we assessed the potential for multicollinearity for explanatory variables using Pearson's correlation coefficient for continuous variables and did not include any variables with an $|r| > 0.60$ in the same model. Following model selection, we further evaluated the potential for multicollinearity using variation inflation factors (VIF) and a cutoff of 10 [46, 51]. We ranked *a priori* models using Akaike's Information Criteria adjusted for small sample sizes and AICc weights [49, 50]. In the event of multiple competing models and model-selection uncertainty, we averaged models that carried >5% AICc weight, consistent with much of the work in natural resources (e.g. [52–61]). We assessed our ranked list of models for any evidence of uninformative parameters and did not include any models with uninformative parameters in model averaging [62, 63].

## Results

Our sample included 384 unique age estimates for 251 individual mule deer (due to recapture in subsequent years for some individuals). The sample was comprised of 6, 42, 64, 79, 107, 81, and 5 deer captured in the years 2014, 2015, 2016, 2017, 2018, 2019, and 2020, respectively. Between 1 and 57 deer were sampled from the 21 management units across Utah with an average of 18 deer sampled per unit. Age estimates based on tooth wear were assigned to 347 females (including recapture events) and 37 males (no recapture events). Our sample was biased toward females given the focus of ongoing efforts to monitor survival and reproduction of females with occasional males collared for collection of survival and migration information.

**Table 2. Model selection table for 19 *a priori* models of age estimates from live animals in the field within one year of cementum age for mule deer (*Odocoileus hemionus*) captured from 2014–2020 in Utah, USA.**

| Model Structure | df[a] | logLik[b] | AICc[c] | Delta[d] | Weight[e] |
|---|---|---|---|---|---|
| Observer+Age | 6 | -150.14 | 312.5 | 0.00 | 0.388 |
| Observer+Age+Sex | 7 | -149.74 | 313.8 | 1.28 | 0.204 |
| Age | 5 | -152.67 | 315.5 | 2.98 | 0.087 |
| *Observer+Age+Sex+Observer*Age | 8 | -149.71 | 315.8 | 3.30 | 0.074 |
| *Observer+Age+Sex+CapturePeriod | 8 | -149.74 | 315.9 | 3.37 | 0.072 |
| Age+Sex | 6 | -152.18 | 316.6 | 4.08 | 0.050 |
| *Observer+Age+CapturePeriod+Observer*Age | 8 | -150.10 | 316.6 | 4.10 | 0.050 |
| *Age+CapturePeriod | 6 | -152.67 | 317.5 | 5.04 | 0.031 |
| *Observer+Age+Sex+CapturePeriod+Obs.*Age | 9 | -149.71 | 317.9 | 5.41 | 0.026 |
| *Age+Sex+CapturePeriod | 7 | -152.18 | 318.7 | 6.16 | 0.018 |
| Observer | 5 | -194.08 | 398.3 | 85.80 | 0.00 |
| Intercept Only | 4 | -195.68 | 399.4 | 86.93 | 0.00 |
| Observer+Sex | 6 | -193.76 | 399.7 | 87.23 | 0.00 |
| *Observer+CapturePeriod | 6 | -194.00 | 400.2 | 87.71 | 0.00 |
| Sex | 5 | -195.38 | 400.9 | 88.39 | 0.00 |
| *Observer+Sex+Observer*Sex | 7 | -193.44 | 401.2 | 88.68 | 0.00 |
| CapturePeriod | 5 | -195.63 | 401.4 | 88.90 | 0.00 |
| *Observer+Sex+CapturePeriod | 7 | -193.70 | 401.7 | 89.19 | 0.00 |
| *Sex+CapturePeriod | 6 | -195.34 | 402.9 | 90.39 | 0.00 |

[a]Degrees of freedom

[b]Log likelihood

[c]Akaike's Information Criteria

[d]AIC relative to the best fitting model

[e]Akaike weight

*Models judged to be uninformative

Models including age of deer and observer experience carried the most AICc weight and these explanatory variables occurred in all top-ranked models (Table 2). We found less support for sex in our modelling effort as it did not occur in the most-supported model (Table 2). Additionally, models with capture period had little to no support, as the highest-ranked model with this variable was fifth with only 7% of the AICc weight (Table 2). Variance inflation factors for supported models were low (1.025–1.032), indicating there was limited correlation between explanatory variables. We judged the interaction with observer and age, the interaction between observer and sex, and also capture period to be uninformative parameters (Table 2).

Due to multiple competing models (models with > 5% weight), we averaged β coefficients from the top four models, excluding any models judged to include uninformative parameters (Table 2). Observer experience influenced accuracy of estimates with experienced observers maintaining higher accuracy than inexperienced observers (model-averaged β estimate -0.8216, 85% CI -1.4885- -0.1547) (Table 3). Cementum age of deer also influenced estimates of accuracy. Accuracy of estimates decreased as cementum age increased (model-averaged β estimate -0.4810, 85% CI -0.5749 –-0.3871) (Table 3). Odds ratios from averaged coefficients showed odds of wear ages matching cementum ages decreased by a factor of 0.4397 for inexperienced observers and 0.6181 for each increase in cementum age (Fig 2). Sex did influence accuracy and was a variable in our second-ranked model (Table 2), however, 85% confidence

**Table 3. Model averages for age estimates within one year of cementum age with an 85% confidence interval for mule deer (*Odocoileus hemionus*) captured from 2014–2020 in Utah, USA.**

| | Estimate[a] | Std. Error[b] | Adj. SE[c] | z-value[d] | LCI[e] | UCI[f] |
|---|---|---|---|---|---|---|
| Intercept | 4.0265 | 0.4669 | 0.4685 | 8.594 | 3.3529 | 4.700 |
| Observer | -0.7975 | 0.3746 | 0.3760 | 2.121 | -1.3381 | -0.2569 |
| Age | -0.4797 | 0.0641 | 0.0643 | 7.455 | -0.5722 | -0.3872 |
| Sex | 0.4044 | 0.5267 | 0.5286 | 0.7650 | -0.3555 | 1.1644 |

[a]Model-averaged β estimate

[b]Standard error

[c]Adjusted R-squared

[d]Number of standard deviations from mean

[e]Lower confidence interval

[f]Upper confidence interval

intervals around the beta estimate overlapped zero (Table 3), suggesting it was not as influential as other variables.

Mean accuracy of estimates within one year of cementum age was 75%. There was a negative relationship between age and accuracy resulting in a decrease in accuracy as deer age increased (Fig 3). The mean bias in accuracy of age estimates was -0.6 years (SE ± 0.086), indicating underestimation of ages of older deer. However, overall bias in estimates was low with mean age estimates falling within one year of cementum age for deer ≤ 8 years old and within two years of cementum age for deer ≤10 years old (Fig 4). Our model-averaged coefficients were influenced by models with an interaction between both cementum age and observer experience. These models suggested that experienced observers had greater accuracy in their age estimates than inexperienced observers did, especially as cementum age.

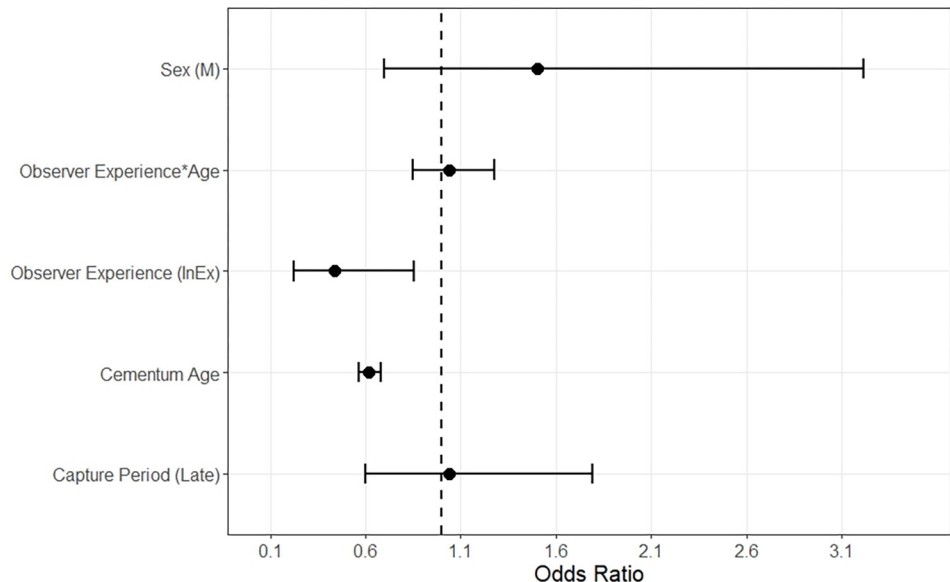

**Fig 2. Odds ratios with standard error bars derived from model-averaged β coefficients of accuracy for age estimates within one year of cementum age for mule deer (*Odocoileus hemionus*) captured from 2014–2020 in Utah, USA.**

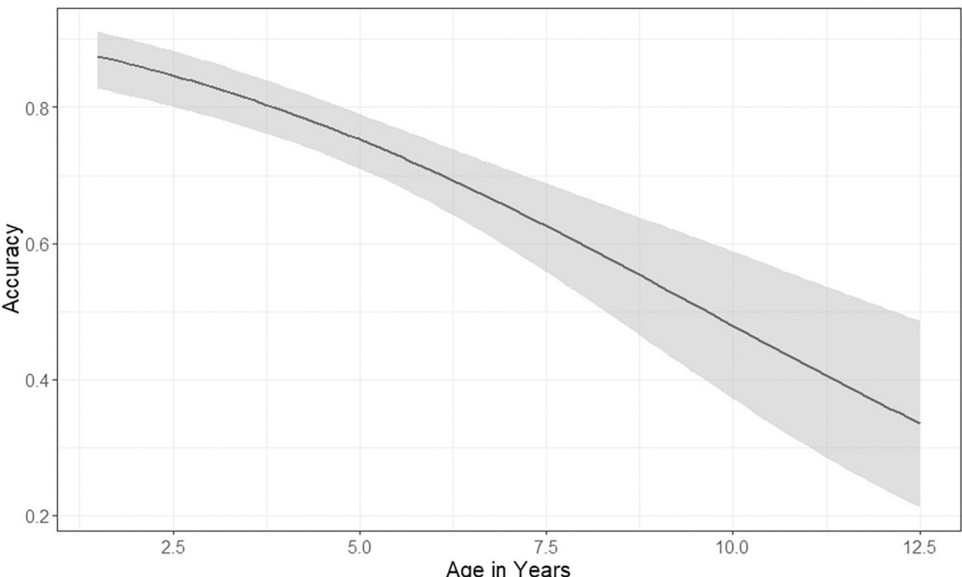

**Fig 3. Accuracy of age estimates with SE bars from live animals in the field within one year of cementum age for mule deer (*Odocoileus hemionus*) captured from 2014–2020 in Utah, USA.** Shaded areas represent an 85% CI.

## Discussion

We observed high accuracy (0.75; SE ± 0.043) and low bias (-0.6 years; SE ± 0.086) using the tooth replacement and wear method to estimate age of mule deer in the field. Our overall estimate of accuracy was biased low indicating inaccuracies were most often due to underestimation, rather than overestimation, of ages. Estimates of age for deer < 4.5 years were largely accurate and unbiased, but accuracy declined linearly with age and old animals (≥ 9 years of age) were consistently underestimated in the field by about 2 years (Fig 4). Decreased accuracy

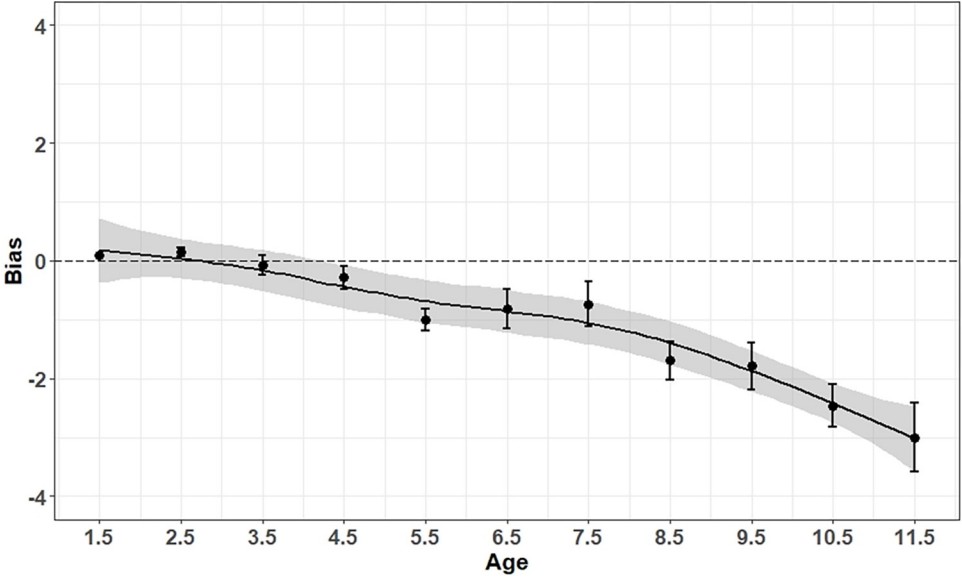

**Fig 4. Bias in accuracy of age estimates with SE bars from live animals in the field (error bars = SE) for mule deer (*Odocoileus hemionus*) captured from 2014–2020 in Utah, USA.** Shaded areas represent an 85% CI.

in estimation of age for older animals is likely due to increased variation in tooth wear for older individuals and fewer available samples as a result of senescence [40]. Although accuracy decreased for older animals, consistent with previous findings, overall accuracy was still high and considered reliable [26, 64, 65].

We found that age estimates made by experienced observers using the tooth replacement and wear method had a higher degree of accuracy than estimates made by observers who were inexperienced. Accuracy of age estimates made by both groups was similar for deer < 4.5 years of age. However, accuracy of estimates for deer > 4.5 years of age by experienced observers remained significantly higher than estimates by inexperienced observers. The difference in accuracy between observers highlights the need for biologists to gain experience–particularly for deer > 4.5 years of age. Nonetheless, because harvest in Utah and many other locations is often composed primarily of male animals < 5.5, it can be challenging to sample an adequate number of older animals. Females tend to be longer-lived than males, but are often not a big percentage of the harvest for mule deer which limits biologists' experience with estimation for older age classes of deer where tooth wear is more variable [66, 67]. Providing observers with adequate experience using the tooth replacement and wear method will ensure higher accuracy of age estimates.

Demography of wildlife populations is strongly influenced by age [68–70]. To understand these demographic rates, however, we first need to understand age structure. Age estimates, when performed by experienced observers, can produce accurate models of age distribution within a population. Broad age categories where all animals over 3 years of age are simply labeled as > 3, however, are less informative—particularly for species like mule deer that regularly live into teenage years [43]. Such a broad categorization is less informative when assessing reproductive potential, habitat selection, disease rates, and especially survival within mule deer herds [71–73]. Because our estimates of age were relatively accurate and only biased low by a couple of years at the oldest age, we recommend use of age estimates over broad categories.

Age estimates, even if biased in accuracy, produce information with greater accuracy than age categories regarding age distribution. Accurate models of age can then be used to make better-informed management decisions [40]. With estimates of age, we can accurately assess life-history traits of mule deer. Age estimates allow for analyses of age-specific survival of deer, movement behavior such as migration and dispersal, and resource selection. With age data, we can ask questions about how age influences reproductive traits including litter size, offspring survival, and frequency of pregnancy. We can also begin to determine whether mule deer experience reproductive senescence or not. We encourage both research institutions and management agencies use patterns in tooth replacement and wear to estimate ages of live mule deer in the field. Furthermore, because of the accuracy (0.75; SE ± 0.043) associated with this method, we suggest the use of age estimates can be confidently used in data analyses surrounding additional demographic rates.

## Conclusions

The results of this study indicate that estimating actual age using patterns in tooth replacement and wear is a viable method. This method is preferable over incisor extraction (cementum analysis) for live deer being released back into the wild. Due to the impact of observer experience on accuracy of estimates, age estimations should be conducted by observers who have significant experience using the tooth replacement and wear method for both sexes and all age cohorts. If an observer is having difficulty choosing an estimate between two ages for an older animal (> 5 years old), it is suggested the observer select the older age due to the observed negative bias in accuracy and frequency of underestimation for older animals that we observed.

Because the tooth replacement and wear method is considered reliable, estimates of age determined during current and future mule deer research efforts should be included as a covariate in future analyses. Including age in these analyses can reveal how and at what rates age influences other demographic rates within mule deer populations.

## Supporting information

**S1 Data.**
(XLSX)

## Acknowledgments

We are grateful to the numerous employees of the Utah Division of Wildlife Resources, past graduate students and undergraduate student employees at Brigham Young University for their contributions to the field and laboratory work associated with this study.

## Author Contributions

**Conceptualization:** Morgan S. Hinton, Randy T. Larsen.

**Data curation:** Morgan S. Hinton.

**Formal analysis:** Morgan S. Hinton, Brock R. McMillan, Randy T. Larsen.

**Funding acquisition:** Brock R. McMillan, Kent R. Hersey, Randy T. Larsen.

**Investigation:** Morgan S. Hinton, Brock R. McMillan, Randy T. Larsen.

**Methodology:** Morgan S. Hinton, Brock R. McMillan, Kent R. Hersey, Randy T. Larsen.

**Project administration:** Randy T. Larsen.

**Resources:** Kent R. Hersey.

**Supervision:** Randy T. Larsen.

**Visualization:** Morgan S. Hinton.

**Writing – original draft:** Morgan S. Hinton.

**Writing – review & editing:** Brock R. McMillan, Kent R. Hersey, Randy T. Larsen.

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
