## [Decision Letter · Decision Letter 0]

17 Oct 2022

PONE-D-22-19575Estimating age of mule deer in the field: Can we move beyond broad age categories?PLOS ONE

Dear Dr. Hinton,

Thank you for submitting your manuscript to PLOS ONE. After careful consideration, we feel that it has merit but does not fully meet PLOS ONE’s publication criteria as it currently stands. Therefore, we invite you to submit a revised version of the manuscript that addresses the points raised during the review process.

Dear Dr. Hinton,

thank you very much for considering PLoS ONE to submit the results of your investigations. I have now received the comments by three independent reviewers on your submission and, while all of them find value on your study, they also point out several aspects to improve before considering your manuscript fully suitable of publication in PLoS ONE. Since the changes propposed by the reviewers imply major modifications in your original submissions, including the suggestion to perform new analyses changing the age category able to be determined through tooth and wear replacement over 2.5 years of age, I must recommend 'major revisions' to be carried out.

I hope you find the comments by the reviewers useful to improve your manuscript and I am looking forward to receive the ammended version of your submission.

Best regards,

We look forward to receiving your revised manuscript.

Kind regards,

Jorge Ramón López-Olvera

Academic Editor

PLOS ONE

Journal Requirements:

“Funding for this research was provided by the American Society of Mammalogists (https://www.mammalsociety.org/), Brigham Young University (https://www.byu.edu/), the Bureau of Land Management (https://www.blm.gov/), the Mule Deer Foundation (https://muledeer.org/), the Rocky Mountain Elk Foundation (https://www.rmef.org/), Safari Club International (https://safariclub.org/), Sportsmen for Fish and Wildlife (https://sfw.net/), the Utah Archery Association (https://www.utaharchery.org/), and the  Utah Division of Wildlife Resources (https://wildlife.utah.gov/). Grant numbers 196313 and 206012 from Utah Division of Wildlife Resources were awarded to BRM and RTL. Additionally, the Utah Division of Wildlife Resources contributed to the data collection associated with this research and the decision to publish it.”  

3. Your ethics statement should only appear in the Methods section of your manuscript. If your ethics statement is written in any section besides the Methods, please delete it from any other section

Additional Editor Comments:

Dear Dr. Hinton,

thank you very much for considering PLoS ONE to submit the results of your investigations. I have now received the comments by three independent reviewers on your submission and, while all of them find value on your study, they also point out several aspects to improve before considering your manuscript fully suitable of publication in PLoS ONE. Since the changes propposed by the reviewers imply major modifications in your original submissions, including the suggestion to perform new analyses changing the age category able to be determined through tooth and wear replacement over 2.5 years of age, I must recommend 'major revisions' to be carried out.

I hope you find the comments by the reviewers useful to improve your manuscript and I am looking forward to receive the ammended version of your submission.

Best regards,

Reviewers' comments:

Reviewer's Responses to Questions

**Comments to the Author**

1. Is the manuscript technically sound, and do the data support the conclusions?

Reviewer #1: Partly

Reviewer #2: Partly

Reviewer #3: Yes

2. Has the statistical analysis been performed appropriately and rigorously? 

Reviewer #1: Yes

Reviewer #2: I Don't Know

Reviewer #3: Yes

3. Have the authors made all data underlying the findings in their manuscript fully available?

Reviewer #1: Yes

Reviewer #2: Yes

Reviewer #3: Yes

4. Is the manuscript presented in an intelligible fashion and written in standard English?

Reviewer #1: Yes

Reviewer #2: Yes

Reviewer #3: Yes

5. Review Comments to the Author

Reviewer #1: This is a well-written manuscript with information that contributes to the literature of aging mule deer, which is arguably growing in importance, given mule deer populations range-wide. Just some minor questions and thought-provoking (hopefully) quibbles.

Lines 68-70; 142-144: The suggestion that aging deer (using the tooth eruption and wear technique [TEW]) at 2.5 years of age is just as accurate as aging deer at 1.5 years of age is misleading and, I would argue, false. At 2.5 years of age, mule deer and white-tailed deer have their permanent (adult) teeth and dental formula, with the only distinguishable features between deer 2.5 years and older deer being tooth wear patterns, which can sometimes be difficult to identify all necessary characteristics in the field on a live individual and can vary based on diet, size of deer, and other factors, as you know. Hamlin et al. (2000), Storm et al. (2014), and Adams & Blanchong (2020) all show error using TEW for aging deer 2.5 years and older.

Hamlin KL, Pac DF, Sime CA, DeSimone RM, Dusek GL. (2000) Evaluating the accuracy of ages obtained by two methods for Montana ungulates. Journal of Wildlife Management. 64,441–449.

Storm DJ, Samuel MD, Rolley RE, Beissel T, Richards BJ, Van Deelen TR. (2014) Estimating ages of white-tailed deer: age and sex patterns of error using tooth wear-and-replacement and consistency of cementum annuli. Wildlife Society Bulletin. 38,849–856.

Adams DM, Blanchong JA (2020) Precision of cementum annuli method for aging male white-tailed deer. PLoS ONE 15(5), e0233421.

Lines 242-244, Fig. 2: The odds ratios (ORs) graphed in Fig. 2 do not match those reported in Lines 242-244. From the beta-estimates, it appears the ORs reported in Lines 242-244 are correct.

Line 281: I’m uncertain that most folks would interpret 75% of TEW-CA age estimate pairings within 1 year of each other as “high accuracy”. That would imply that, using TEW aging in the field, managers would be assigning an incorrect age to greater than 25% of deer they’re aging 2.5 years and older. Depending on how different age-specific demographic parameters are from year-to-year, the potential for error could be quite large. While CA aging is not perfectly accurate either, as showed by those cited above as well as Asmus & Weckerly (2002) and Veiberg et al. (2020), CA aging is much more accurate than TEW, especially at older age classes. That’s why most recommend that if managers need ages accurate to the year for age-specific demographic parameters, that CA aging is the best bet.

Some other questions/thoughts: it would be nice to see a table of the raw TEW estimates vs the CA age estimates to see the "unmodeled" accuracy. Did you explore categories other than 3.5 and older (e.g., 2-4 years, 5-7 years, 8 years and older [not an actual suggestion, just an example, more thought would need to be given to appropriate age categories based on demographic parameters) and if those may be more useful to managers? Key to all of this is do demographic parameters for deer change at different ages? That is where age categories of importance could be identified. Maybe the answer to the question in your title is somewhere in between the 2.5 years and older category of old and the single-year category.

Reviewer #2: This manuscript evaluates the accuracy of tooth wear and replacement patterns for aging mule deer in Utah by comparing it to cementum analysis. Age information is often important for wildlife population management including estimating population demographic rates. I think the manuscript would be more useful if the authors revised it to increase specificity with respect to the background information and objectives, offer more specific detail about the data collected and how they were used in analysis, and provide a more careful discussion of the interpretation and implications of the findings that brings in the vast literature on this topic.

At the moment, the manuscript lacks important detail in numerous places that makes it difficult to evaluate the Results or the relatively strong conclusions/ recommendations provided in the Discussion and Conclusions sections. Further, in the Introduction there are generalizations made about aging using teeth that are not appropriate. These statements should be revised to more specifically speak to the taxa of interest in the study. The section is also under referenced. The Discussion is extremely under-referenced. Finally, the conclusions and recommendations are overly strong and applied too broadly relative to the data presented and without adequate evidence to support them from the literature (references). I provide specific line-by-line comments below.

Line 53-45 – Sex is not easy to determine in the wild for numerous wildlife species. So, being more specific about which taxa you are referring to would be useful.

Line 62-63 – Reference needed.

Line 66 - Clarify you mean from live animals.

Line 67-70 - This citation is for fallow deer. Is it appropriate to make such a broad generalization based on this one reference for one species?

Line 71-72- Reference needed.

Line 73-76 - You are talking about cervids here so be specific. Again, this sentence is making a broad generalization when I don't think you mean to.

Line 85-88 - Again, you are not doing this to generalize to all wildlife (or if that is your intent, I think you should not)...you have a particular taxon of focus and you should state that.

Line 91 – Clarify what you mean by “field aging” = using tooth eruption and wear patterns. Also, it should be noted that aging based on cementum annuli is not perfect. There is literature about this. So, you should be careful using it as a “true age”.

Line 133-134 – What is your basis for 200 deer as the cutoff for experienced vs inexperienced? Justify. How many people did you have in each category? What was the distribution of experience in each category? How many agers in total did you end up having in each category?

Line 134-135 - This is an awkward arrangement. Why talk about 1.5 to 2.5 year olds (without describing how) and then talk about the younger animals (fawns) and then go back to the older animals?

Line 187-188 - What does within 1 year mean in terms of what your data were? It's not clear what the age categories are for TWR vs cementum that you are using so I can't tell if within 1 year means the same age or something different. Perhaps you could provide a table with more of the actual data rather than moving straight to outputs of regression models so the reader can see for themselves exactly what you were working with.

Line 207-208 - Justify this. Model averaging is not universally agreed upon as an appropriate strategy. Also justify the choice of 5% with a reference.

Line 208-211- Why did you do this? And what was your criterion for uninformative parameters? I also don't understand what "models occurred with other competing models" means.

Line 222-223 - Your second model was within 2 AIC of your top model. The standard is generally that any model within 2 AIC of the top model is competitive. So, justify why sex is not important.

Table 1 – Is “1” the intercept only model? If so, make that clear.

Line 245-247 - Confusing as written. What age is most accurate for males and how do you know?

Table 2 - Why would you choose an 85% CI?

Fig 2. – I think this figure is mislabeled. Your experience category has error bars above and below an odds ratio of 1 (1 being no association). This does not make sense in terms of being significant. And, capture period is totally below 1 suggesting it is significant - which is not reflected in Table 2. Also, indicate in the legend what the error bars are.

Fig. 3-5 – What are the error bars and envelopes (shaded areas) on these figures?

Fig. 5 – How many deer had to be aged for someone to qualify as experienced? Methods says 200. Here 200 of each sex – so 400 total? This is unclear.

Line 280-281 - Clarify what your definition of reliable is.

Paragraph 1 (line 280-291) - I think you need to go into what this level of "reliability" pertains to. What management contexts would the level of accuracy you found be sufficient and in what contexts/questions would more accurate aging be needed? This is important and should be explored more deeply and use the literature.

Paragraph 2 (line 296-307) - This all seems really speculative and you've provided no assessment of accuracy based on experience for males so you really can't support your statements about sex based on what you’ve written here. Also, you need references to support the general statements you are making in lines 301-307.

Paragraph 3 (308-316) and 4 (line317-328) – Again, the statements in these paragraphs need references to support them.

Line 308 – What does this sentence mean?

Line 326-428 - Your "high" is subjective. Be specific about what you found.

Line 334-336 – I think this is a risky recommendation to make for all mule deer.

Reviewer #3: Hinton et al. presented a ms entitled “Estimating age of mule deer in the field: Can we move beyond broad age categories?” for publication in PLOS ONE. The paper deals with a relevant and interesting topic: age estimation (teeth eruption/wearing) versus age determination (analysis of cementum annuli from teeth) in ungulates. The ms reported the case of mule deer (Odocoileus hemionous; sample size: 384 unique age estimates for 251 individuals, 347 females and 37 males) aging across the State of Utah.

I enjoyed reading this ms and I think that this study is a valuable contribution to improve the knowledge/management of this species.

However, despite the interesting topic and the potential of management implications for the target species, I believe that a revision is required (please, see my comments below).

General comments:

1) A first flaw that I found concerns the absence of an accurate description of the age estimation method. The conclusions of this ms open up the possibility of using age estimation in future analyzes but do not describe how ages were estimated.

The Authors refer to two old studies and a book (Severinghaus 1949, Robinette 1957, Heffelfinger 2006). However, the experience in age estimation resulted an important variable.

I believe this is a fundamental point: the Authors must describe how ages were estimated by expert biologists through the use of detailed descriptions and photographs. Only in this way their findings can be useful to the scientific community.

2) The sample of males is very low. I suggest to carry out the analyzes separately for the two sexes.

3) Is the age of these animals estimated using the same criteria for males and females?

4) I believe it is important to compare animals in which age was estimated by tooth eruption stages and those in which it was evaluated by tooth wear. I have no direct experience on this species but for many other ungulates species the estimation of age by eruption is very different from the wear processes. Is the experience important also using eruption stages?

5) Another very important issue is related to the use of animals coming from very different areas (21 management units within five administrative regions of the Utah Division of Wildlife Resources). These areas have very different altitudinal ranges, vegetation communities, etc etc…

The Authors should take into account the “data collection area” in their models, not only as a random factor (e.g., altitudinal range, vegetation, …). This analysis could help to understand if the underestimation is the same over the whole large study area or if this is true only for certain areas (for example areas with lower altitudes).

6. PLOS authors have the option to publish the peer review history of their article (what does this mean?). If published, this will include your full peer review and any attached files.

Reviewer #1: No

Reviewer #2: No

Reviewer #3: No

---

## [Author Response · Author response to Decision Letter 0]

9 Jan 2023

EDITOR COMMENTS:

-We have reformatted the manuscript and figures to meet PLOS ONE’s style requirements.

 -We have amended the Role of Funder Statement and included it in the cover letter.

3. Your ethics statement should only appear in the Methods section of your manuscript. If your ethics statement is written in any section besides the Methods, please delete it from any other section

-Ethics statement has been moved to the Methods section and removed from the end of the document.

 -Figure has been edited to remove copyrighted satellite imagery from map.

REVIEWER COMMENTS

Reviewer #1: This is a well-written manuscript with information that contributes to the literature of aging mule deer, which is arguably growing in importance, given mule deer populations range-wide. Just some minor questions and thought-provoking (hopefully) quibbles.

Lines 68-70; 142-144: The suggestion that aging deer (using the tooth eruption and wear technique [TEW]) at 2.5 years of age is just as accurate as aging deer at 1.5 years of age is misleading and, I would argue, false. At 2.5 years of age, mule deer and white-tailed deer have their permanent (adult) teeth and dental formula, with the only distinguishable features between deer 2.5 years and older deer being tooth wear patterns, which can sometimes be difficult to identify all necessary characteristics in the field on a live individual and can vary based on diet, size of deer, and other factors, as you know. Hamlin et al. (2000), Storm et al. (2014), and Adams & Blanchong (2020) all show error using TEW for aging deer 2.5 years and older.

 -We are a little confused by this comment because Figure 3 shows decreased accuracy for 2.5 year-old deer compared to 1.5 year-old deer. For both ages, accuracy was high, however, the point estimate for accuracy of 2.5 year olds is lower. Does the comment refer to Figure 4 regarding bias? Language in the introduction and methods sections has been added to provide clarity. 

Lines 242-244, Fig. 2: The odds ratios (ORs) graphed in Fig. 2 do not match those reported in Lines 242-244. From the beta-estimates, it appears the ORs reported in Lines 242-244 are correct.

 -We are also confused by this comment. The estimates in lines 242-244 are beta estimates on the logit scale as shown in Table 2. Figure 2 reports odds ratios which are easily calculated from the beta estimates. We’ve adjusted the language in the caption for Figure 2 to more clearly articulate that the figure refers to odds ratios. This change should clarify for the reader. 

Line 281: I’m uncertain that most folks would interpret 75% of TEW-CA age estimate pairings within 1 year of each other as “high accuracy”. That would imply that, using TEW aging in the field, managers would be assigning an incorrect age to greater than 25% of deer they’re aging 2.5 years and older. Depending on how different age-specific demographic parameters are from year-to-year, the potential for error could be quite large. While CA aging is not perfectly accurate either, as showed by those cited above as well as Asmus & Weckerly (2002) and Veiberg et al. (2020), CA aging is much more accurate than TEW, especially at older age classes. That’s why most recommend that if managers need ages accurate to the year for age-specific demographic parameters, that CA aging is the best bet.

-We agree that CA is the most accurate method for aging ungulates. This method, however, is invasive and impractical for a live animal being released back into the wild (Festa-Bianchet 2002). Following suggestions from reviewer 2, we have added the term “live” to highlight the TRW method being the most commonly used method for live animals.

Some other questions/thoughts: it would be nice to see a table of the raw TEW estimates vs the CA age estimates to see the "unmodeled" accuracy. Did you explore categories other than 3.5 and older (e.g., 2-4 years, 5-7 years, 8 years and older [not an actual suggestion, just an example, more thought would need to be given to appropriate age categories based on demographic parameters) and if those may be more useful to managers? Key to all of this is do demographic parameters for deer change at different ages? That is where age categories of importance could be identified. Maybe the answer to the question in your title is somewhere in between the 2.5 years and older category of old and the single-year category.

 -The raw data for ages estimated by TRW compared to estimates derived from CA for each individual are available for this manuscript. We agree that more specific age categories as mentioned could be beneficial to managers than 3.5+, however, such categories are not yet suggested in the literature. However, categories cannot precisely indicate at which ages demographic parameters change for deer. If such categories are to be determined, further research needs to be conducted to indicate the ages at which demography of mule deer changes. 

Reviewer #2: This manuscript evaluates the accuracy of tooth wear and replacement patterns for aging mule deer in Utah by comparing it to cementum analysis. Age information is often important for wildlife population management including estimating population demographic rates. I think the manuscript would be more useful if the authors revised it to increase specificity with respect to the background information and objectives, offer more specific detail about the data collected and how they were used in analysis, and provide a more careful discussion of the interpretation and implications of the findings that brings in the vast literature on this topic.

At the moment, the manuscript lacks important detail in numerous places that makes it difficult to evaluate the Results or the relatively strong conclusions/ recommendations provided in the Discussion and Conclusions sections. Further, in the Introduction there are generalizations made about aging using teeth that are not appropriate. These statements should be revised to more specifically speak to the taxa of interest in the study. The section is also under referenced. The Discussion is extremely under-referenced. Finally, the conclusions and recommendations are overly strong and applied too broadly relative to the data presented and without adequate evidence to support them from the literature (references). I provide specific line-by-line comments below.

-Thank you for the comments. Taxa clarity was added both in wording and in references. Additional references were added to both the introduction and discussion. 

Line 53-45 – Sex is not easy to determine in the wild for numerous wildlife species. So, being more specific about which taxa you are referring to would be useful.

-Changed to Cervids.

Line 62-63 – Reference needed.

-References added to support the sentence. 

Line 66 - Clarify you mean from live animals.

-Changed with reviewer recommendation

Line 67-70 - This citation is for fallow deer. Is it appropriate to make such a broad generalization based on this one reference for one species?

-Additional citations added and language changed to provide clarity. 

Line 71-72- Reference needed.

-References added to the sentence.

Line 73-76 - You are talking about cervids here so be specific. Again, this sentence is making a broad generalization when I don't think you mean to.

-Changed to reviewer recommendation, specified cervids.

Line 85-88 - Again, you are not doing this to generalize to all wildlife (or if that is your intent, I think you should not)...you have a particular taxon of focus and you should state that.

-Changed to reviewer recommendation, specified cervids

Line 91 – Clarify what you mean by “field aging” = using tooth eruption and wear patterns. Also, it should be noted that aging based on cementum annuli is not perfect. There is literature about this. So, you should be careful using it as a “true age”.

-Changed with reviewer recommendation to “tooth replacement and wear method” and changed “true age” to cementum age. 

Line 133-134 – What is your basis for 200 deer as the cutoff for experienced vs inexperienced? Justify. How many people did you have in each category? What was the distribution of experience in each category? How many agers in total did you end up having in each category?

-We have added wording to this section to justify this using this number as a cutoff.

Line 134-135 - This is an awkward arrangement. Why talk about 1.5 to 2.5 year olds (without describing how) and then talk about the younger animals (fawns) and then go back to the older animals?

-Changed to reviewer recommendation, changed sentence arrangement to follow ages

Line 187-188 - What does within 1 year mean in terms of what your data were? It's not clear what the age categories are for TWR vs cementum that you are using so I can't tell if within 1 year means the same age or something different. Perhaps you could provide a table with more of the actual data rather than moving straight to outputs of regression models so the reader can see for themselves exactly what you were working with.

-We aged each animal to the year for both CA and TWR (age categories = “1.5, 2.5, 3.5. etc.). We’ve adjusted the wording of lines in the data analysis section to be more clear about the meaning of within 1 year. 

Line 207-208 - Justify this. Model averaging is not universally agreed upon as an appropriate strategy. Also justify the choice of 5% with a reference.

-References have been added to support the choice of 5%.

Line 208-211- Why did you do this? And what was your criterion for uninformative parameters? I also don't understand what "models occurred with other competing models" means.

-Thanks for the note. We have added a reference and rewritten the sentence to clarify.

Line 222-223 - Your second model was within 2 AIC of your top model. The standard is generally that any model within 2 AIC of the top model is competitive. So, justify why sex is not important.

-Yes. We model averaged given our results showed multiple competing models. The confidence intervals around the beta estimate for sex, however, overlapped zero suggesting it was not as influential as other variables. We’ve adjusted the language to clarify. 

Table 1 – Is “1” the intercept only model? If so, make that clear.

-Changed to Intercept Only in the table.

Line 245-247 - Confusing as written. What age is most accurate for males and how do you know?

-Reworded for clarity.

Table 2 - Why would you choose an 85% CI?

-Because model selection uses a “weight of evidence” approach as opposed to an up or down vote based on a p-value, recommendations now suggest using 85% confidence intervals around beta estimates to assess direction and strength of effect sizes (see Arnold 2010).

Fig 2. – I think this figure is mislabeled. Your experience category has error bars above and below an odds ratio of 1 (1 being no association). This does not make sense in terms of being significant. And, capture period is totally below 1 suggesting it is significant - which is not reflected in Table 2. Also, indicate in the legend what the error bars are.

-Thank you for catching this. The figure labels have been corrected to the appropriate ones, which align with the data shown in Table 2. Figure caption has been changed to denote SE bars.

Fig. 3-5 – What are the error bars and envelopes (shaded areas) on these figures?

-Added language to figure legends to indicate SE bars and confidence intervals. 

Fig. 5 – How many deer had to be aged for someone to qualify as experienced? Methods says 200. Here 200 of each sex – so 400 total? This is unclear.

-Thank you for catching this. We have corrected this in the figure caption to reflect the correct number of animals. 

Line 280-281 - Clarify what your definition of reliable is.

-Thank you for this comment. Language was added to the sentence for clarity. 

Paragraph 1 (line 280-291) - I think you need to go into what this level of "reliability" pertains to. What management contexts would the level of accuracy you found be sufficient and in what contexts/questions would more accurate aging be needed? This is important and should be explored more deeply and use the literature.

-Thank you for this comment. The management contexts associated with the accuracy found in this study are reported, with citations, in lines 321-330.

Paragraph 2 (line 296-307) - This all seems really speculative and you've provided no assessment of accuracy based on experience for males so you really can't support your statements about sex based on what you’ve written here. Also, you need references to support the general statements you are making in lines 301-307.

-Paragraph has been rewritten for clarity. References have been added to these lines. 

Paragraph 3 (308-316) and 4 (line317-328) – Again, the statements in these paragraphs need references to support them.

-References have been added to these lines.

Line 308 – What does this sentence mean?

-Reworded and changed language for clarity.

Line 326-428 - Your "high" is subjective. Be specific about what you found.

-We agree that the term high is subjective. We removed the word. 

Line 334-336 – I think this is a risky recommendation to make for all mule deer.

-Thank you for the comment. We reworded our conclusions to more accurately reflect the usefulness of this method. 

Reviewer #3: Hinton et al. presented a ms entitled “Estimating age of mule deer in the field: Can we move beyond broad age categories?” for publication in PLOS ONE. The paper deals with a relevant and interesting topic: age estimation (teeth eruption/wearing) versus age determination (analysis of cementum annuli from teeth) in ungulates. The ms reported the case of mule deer (Odocoileus hemionous; sample size: 384 unique age estimates for 251 individuals, 347 females and 37 males) aging across the State of Utah.

I enjoyed reading this ms and I think that this study is a valuable contribution to improve the knowledge/management of this species.

However, despite the interesting topic and the potential of management implications for the target species, I believe that a revision is required (please, see my comments below).

General comments:

1) A first flaw that I found concerns the absence of an accurate description of the age estimation method. The conclusions of this ms open up the possibility of using age estimation in future analyzes but do not describe how ages were estimated.

The Authors refer to two old studies and a book (Severinghaus 1949, Robinette 1957, Heffelfinger 2006). However, the experience in age estimation resulted an important variable.

I believe this is a fundamental point: the Authors must describe how ages were estimated by expert biologists through the use of detailed descriptions and photographs. Only in this way their findings can be useful to the scientific community.

-Thank you for this comment. We have added a table in the methods section outlining the general dental characteristics associated with each age of mule deer. 

2) The sample of males is very low. I suggest to carry out the analyzes separately for the two sexes.

-The results show that sex was not a significant variable in the model selection process, indicating that there is not a significant difference in age estimates for males vs. females. 

3) Is the age of these animals estimated using the same criteria for males and females?

-Yes. An additional sentence was added to clarify the same criteria was used to estimate age of both sexes.

4) I believe it is important to compare animals in which age was estimated by tooth eruption stages and those in which it was evaluated by tooth wear. I have no direct experience on this species but for many other ungulates species the estimation of age by eruption is very different from the wear processes. Is the experience important also using eruption stages?

-These are two different processes used to estimate two different age classes of animals. Estimation using tooth eruption is only applicable to deer 2.5 years old and younger and estimation using tooth wear is only applicable to deer 3.5 years old and greater. 

5) Another very important issue is related to the use of animals coming from very different areas (21 management units within five administrative regions of the Utah Division of Wildlife Resources). These areas have very different altitudinal ranges, vegetation communities, etc etc…

The Authors should take into account the “data collection area” in their models, not only as a random factor (e.g., altitudinal range, vegetation, …). This analysis could help to understand if the underestimation is the same over the whole large study area or if this is true only for certain areas (for example areas with lower altitudes).

-Analyses were performed to determine the effect of unit of capture and ecoregion on accuracy of age estimates. Results indicated there was no significant influence of either variable on accuracy of estimates. We added a sentence in the methods section to summarize this finding.

---

## [Decision Letter · Decision Letter 1]

15 Feb 2023

PONE-D-22-19575R1Estimating age of mule deer in the field: Can we move beyond broad age categories?PLOS ONE

Dear Dr. Hinton,

Thank you for submitting your manuscript to PLOS ONE. After careful consideration, we feel that it has merit but does not fully meet PLOS ONE’s publication criteria as it currently stands. Therefore, we invite you to submit a revised version of the manuscript that addresses the points raised during the review process.

Dear Dr. Hinton, I have now received the comments on the revised version of your manuscript by two of the three reviewers who assessed the first submission and, while both think that your manuscript has improved and approaches acceptance, one of them still pointed out several issues which should be addressed before acceptance. I therefore recommend you to undertake the required amendments. I am looking forward to receiving the new version of your manuscript.

We look forward to receiving your revised manuscript.

Kind regards,

Jorge Ramón López-Olvera

Academic Editor

PLOS ONE

Journal Requirements:

Additional Editor Comments (if provided):

Dear Dr. Hinton,

I have now received the comments on the revised version of your manuscript by two of the three reviewers who assessed the first submission and, while both think that your manuscript has improved and approaches acceptance, one of them still pointed out several issues which should be addressed before acceptance.

I therefore recommend you to undertake the required amendments. I am looking forward to receiving the new version of your manuscript.

Reviewers' comments:

Reviewer's Responses to Questions

**Comments to the Author**

1. If the authors have adequately addressed your comments raised in a previous round of review and you feel that this manuscript is now acceptable for publication, you may indicate that here to bypass the “Comments to the Author” section, enter your conflict of interest statement in the “Confidential to Editor” section, and submit your "Accept" recommendation.

Reviewer #1: (No Response)

Reviewer #3: All comments have been addressed

2. Is the manuscript technically sound, and do the data support the conclusions?

Reviewer #1: Partly

Reviewer #3: Yes

3. Has the statistical analysis been performed appropriately and rigorously? 

Reviewer #1: No

Reviewer #3: Yes

4. Have the authors made all data underlying the findings in their manuscript fully available?

Reviewer #1: (No Response)

Reviewer #3: Yes

5. Is the manuscript presented in an intelligible fashion and written in standard English?

Reviewer #1: Yes

Reviewer #3: Yes

6. Review Comments to the Author

Reviewer #1: The manuscript was improved from its previous version with statements that added clarity. However, after further review, I have concerns about the statistical analyses and the subsequent presentation/interpretation of those results.

Line 140: “…to identify deer 1.5 years old from deer older than 1.5 years.”

Table 1: This table is unnecessary as it is described in your citations [24-25,40]. Additionally, it insinuates that it is a straight-forward method, which it’s known that it isn’t, and that the authors point out in Lines 299-301 (“…due to increased variation in tooth wear for older individuals…”). I would just leave it as it is in Line 148 that older deer were aged based on wear patterns.

Lines 217-220: The sentences beginning on Line 217 and Line 219 effectively say the same thing. Combine the two.

Lines 219-220: Although I don’t agree with the model-averaging technique used, if the statement of “much of the work in natural resources” is to be used, more citations would be necessary.

Lines 220-221: If the models were assessed for evidence of uninformative parameters, why were they still included them in the final model?

Lines 234-235: Again, if a parameter wasn’t statistically supported in any of the models, why was it still included in the final model?

Lines 252-254: Typically, if there’s a significant interaction term included in a model, the two coefficients (Observer & Age, in this case) are nearly meaningless by themselves, as they would be dependent on the other. However, since the interaction term itself is nearly 0 and insignificant, I suppose these are fine. Although I’d again question the inclusion of these terms that are nearly 0. We strive to find the simplest model that explains the most about our data (i.e., the basis of AIC), so why include all these uninformative parameters when two models with 3 different predictor variables were found that best describe the data?

Fig. 2: Why is the interaction term not included when all other variables are?

Figs. 3 & 5: One of these is unnecessary. Going back to the comments about Lines 252-254: typically, Fig. 3 would not be warranted because the plot of the interaction would illustrate the two variables. But since the interaction is uninformative, the interaction plot isn’t needed either.

Throughout the paper: Choose and stick with one of the commonly used phrases of the “tooth eruption and wear method”, as it is inconsistent throughout (e.g., TRW, TEW, etc.). Line 315 uses “dental wear and eruption method” and Lines 340-341 use “tooth wear and replacement” method, for example.

Reviewer #3: (No Response)

7. PLOS authors have the option to publish the peer review history of their article (what does this mean?). If published, this will include your full peer review and any attached files.

Reviewer #1: No

Reviewer #3: No

---

## [Author Response · Author response to Decision Letter 1]

30 Mar 2023

Reviewer #1: The manuscript was improved from its previous version with statements that added clarity. However, after further review, I have concerns about the statistical analyses and the subsequent presentation/interpretation of those results.

Line 140: “…to identify deer 1.5 years old from deer older than 1.5 years.”

- Changed to reviewer recommendation.

Table 1: This table is unnecessary as it is described in your citations [24-25,40]. Additionally, it insinuates that it is a straight-forward method, which it’s known that it isn’t, and that the authors point out in Lines 299-301 (“…due to increased variation in tooth wear for older individuals…”). I would just leave it as it is in Line 148 that older deer were aged based on wear patterns.

- This table was added to the manuscript per request of Reviewer #3: “I believe this is a fundamental point: the Authors must describe how ages were estimated by expert biologists through the use of detailed descriptions and photographs. Only in this way their findings can be useful to the scientific community.” Lines 152-154 caution readers “These characteristics are not absolutes, but rather general guidelines that have been supported by frequent feedback determined from cementum analysis.”

Lines 217-220: The sentences beginning on Line 217 and Line 219 effectively say the same thing. Combine the two.

- Thank you for the comment. Sentences were combined as the reviewer suggested. 

Lines 219-220: Although I don’t agree with the model-averaging technique used, if the statement of “much of the work in natural resources” is to be used, more citations would be necessary.

- Added several more citations of studies using model-averaging to this statement.

Lines 220-221: If the models were assessed for evidence of uninformative parameters, why were they still included them in the final model?

- Thank you for catching this. Models judged to be informative were removed from the final model and noted in the model list as uninformative. 

Lines 234-235: Again, if a parameter wasn’t statistically supported in any of the models, why was it still included in the final model?

- Same as reply above, uninformative models were removed from final model. 

Lines 252-254: Typically, if there’s a significant interaction term included in a model, the two coefficients (Observer & Age, in this case) are nearly meaningless by themselves, as they would be dependent on the other. However, since the interaction term itself is nearly 0 and insignificant, I suppose these are fine. Although I’d again question the inclusion of these terms that are nearly 0. We strive to find the simplest model that explains the most about our data (i.e., the basis of AIC), so why include all these uninformative parameters when two models with 3 different predictor variables were found that best describe the data?

- Same as reply above, uninformative models were removed from final model. 

Fig. 2: Why is the interaction term not included when all other variables are?

- Interaction term added to this figure. Thank you for the comment.

Figs. 3 & 5: One of these is unnecessary. Going back to the comments about Lines 252-254: typically, Fig. 3 would not be warranted because the plot of the interaction would illustrate the two variables. But since the interaction is uninformative, the interaction plot isn’t needed either.

- Thank you for this feedback. Removed Figure 5 from manuscript, as the age and observer interaction was found to be uninformative. 

Throughout the paper: Choose and stick with one of the commonly used phrases of the “tooth eruption and wear method”, as it is inconsistent throughout (e.g., TRW, TEW, etc.). Line 315 uses “dental wear and eruption method” and Lines 340-341 use “tooth wear and replacement” method, for example.

- Thank you for catching this. All terminology for this method has been changed to “tooth replacement and wear method” to create consistency and align with the most commonly used term for the method.

---

## [Editor Report · Decision Letter 2]

4 Apr 2023

Estimating age of mule deer in the field: Can we move beyond broad age categories?

PONE-D-22-19575R2

Dear Dr. Hinton,

We’re pleased to inform you that your manuscript has been judged scientifically suitable for publication and will be formally accepted for publication once it meets all outstanding technical requirements.

Kind regards,

Jorge Ramón López-Olvera

Academic Editor

PLOS ONE

Additional Editor Comments (optional):

Dear Dr. Hinton,

thank you for submitting the second revision of your manuscript. Since the last reviewer raising concerns about your submission pointed out just really minor changes, it has been easy to check and verify that they have been satisfactorily carried out, so I am happy to recommend your manuscript for acceptance to be published in PLoS ONE.

Best regards,
---

## [Editor Report · Acceptance letter]

14 Apr 2023

PONE-D-22-19575R2 

Estimating age of mule deer in the field: Can we move beyond broad age categories? 

Dear Dr. Hinton:

I'm pleased to inform you that your manuscript has been deemed suitable for publication in PLOS ONE. Congratulations! Your manuscript is now with our production department. 

Kind regards, 

on behalf of

Dr. Jorge Ramón López-Olvera 

Academic Editor

PLOS ONE